# *Bombyx mori Vps13d* is a key gene affecting silk yield

**Luochao Zhao**[1☯]**, Xia Sun**[1,2☯]**, Xueyang Wang**[1,2☯]**, Sheng Qin**[1,2☯]**, Yunhui Kong**[1,2☯]**, Muwang Li**[1,2☯]*****

1 Jiangsu Key Laboratory of Sericultural Biology and Biotechnology, School of Biotechnology, Jiangsu University of Science and Technology, Zhenjiang, 212018, Jiangsu, China, 2 The Key Laboratory of Silkworm and Mulberry Genetic Improvement, Ministry of Agriculture, Sericultural Research Institute, Chinese Academy of Agricultural Science, Zhenjiang, 212018, Jiangsu, China

☯ These authors contributed equally to this work.
* mwli@just.edu.cn

## Abstract

*Bombyx mori* is an important economic insect, its economic value mainly reflected in the silk yield. The major functional genes affecting the silk yield of *B. mori* have not been determined yet. *Bombyx mori vacuolar protein sorting-associated protein 13d* (*BmVps13d*) has been identified, but its function is not reported. In this study, BmVps13d protein shared 30.84% and 34.35% identity with that of in *Drosophila melanogaste*r and *Homo. sapiens*, respectively. The expressions of *BmVps13d* were significantly higher in the midgut and silk gland of JS (high silk yield) than in that of L10 (low silk yield). An insertion of 9 bp nucleotides and two deficiencies of adenine ribonucleotides in the putative promoter region of *BmVps13d* gene in L10 resulted in the decline of promoter activity was confirmed using dual luciferase assay. Finally, the functions of *BmVps13d* in *B. mori* were studied using the CRISPR/Cas9 system, and the mutation of *BmVps13d* resulted in a 24.7% decline in weight of larvae, as well as a 27.1% (female) decline and a 11.8% (male) decline in the silk yield. This study provides a foundation for studying the molecular mechanism of silk yield and breeding the silkworm with high silk yield.

## Introduction

High silk yield is one of the most important parameters in silkworm breeding. The silk yield is controlled by multiple genes and is among the quantitative traits [1]. Therefore, it is very necessary to study the functional genes that affect the silk yield in *B. mori*. It has been reported that the functional genes in *B. mori* affecting the silk yield have been studied in the level of genes expression preliminarily. In 2015, compared to *Bombyx mandarina*, 16 up-regulated genes related to protein secretion, tissue development, and energy metabolism were identified in the silk gland transcriptome of *B. mori* at the third day of fifth instar larval stage [2]. *BGIBMGA003330* and *BGIBMGA005780* encoding uridine nucleosidase and small heat shock protein respectively were significantly related to cocoon shell weight by comparing the transcriptome of silkworm varieties with different silk yields, which suggested that they played a

**Data Availability Statement:** All relevant data are within the paper and its Supporting Information files.

**Funding:** This project was supported by the National Natural Science Foundation of China

(grant No.32072790) and the Postgraduate Research & Practice Innovation Program of Jiangsu Province (KYCX21_3514). The funders had no role in study design, data collection and analysis, decision to publish, or preparation of the manuscript.

**Competing interests:** The authors declare that they have no competing interests.

role in silkworm silk gland development and silk protein synthesis [3]. Recently, Fang *et al.* confirmed that *KWMTBOMO04917* encoding 5-aminolevulinate synthase (*BmALAS*) and *KWMTBOMO12906* encoding DNA polymerase epsilon (Pol ε) subunit 4 (*POLE4*) in *B. mori* were highly expressed by quantitative real-time polymerase chain reaction (RT-qPCR) and were positively related to synthesis of silk proteins [4]. Although many studies have provided some insights into the genetic basis of the silk yield, the major functional genes have not been determined yet, and the mechanism of affecting cocoon characters is still unclear.

Silk protein is synthesized in silk gland and consists of fibroin and sericin proteins in *B. mori* [5]. The synthesis of silk protein is associated with the expression of the genes related to bioenergy metabolism, ribosome, and protein transport [1]. It has been studied that vacuolar protein sorting (VPS) proteins belong to the endosomal sorting complex required for transport (ESCRT) and have the functions of protein transport and sorting [6]. They are necessary for sorting ubiquitinated membrane proteins into the multivesicular body (MVB) by deforming them inward and then breaking them to release the newly formed vesicles [7,8]. *BmVps13d* was identified from the whole genome sequencing of 137 representative silkworm strains with different cocoon characters [9]. It has been reported that *Vps13d* plays an extremely important role of mild intellectual development and energy metabolism. Cognitive impairment ranging from mild intellectual disability to developmental delay when the function of *Vps13d* is discorded in humans [10]. *Vps13d* is an essential gene that is necessary for energy metabolism, growth and development in *D. melanogaster*, and the loss of this gene affects normal development and even leads it to death. [11]. In HeLa cells, *Vps13d*-KO exhibits abnormal mitochondrial morphology and leads to either partial or complete peroxisome loss in several transformed cell lines and fibroblasts [12].

In this study, to figure out the functions of *BmVps13d* in *B. mori*, the sequence differences and expression levels of *BmVps13d* in JS and L10 were analyzed first, and then the *BmVps13d* promoter activity was confirmed. Finally, the *BmVps13d* mutant *B. mori* strain was constructed. The results provide some new insights into the mechanism for the improvement of silkworm strains by molecular means and promote superior silkworm strain breeding.

## Materials and methods

### Silkworm culture and sample preparation

Three silkworm strains, Nistari, JS, and L10 used in this study were provided by the National Sericulture Resources Conservation Center, Chinese Academy of Agricultural Sciences. The Cas9 transgenic silkworms with green fluorescence were retained in our laboratory. The larvae were fed with fresh mulberry leaves at 25˚C relative humidity of 72–78%, and cultured under 12 h light and 12 h dark conditions. Midgut, silk gland, and fat body samples were dissected from JS and L10 at the third day of fifth instar larval stage. Larva samples were collected at the beginning and end of each instar respectively. The pupae were taken every 48 h from the first day to the eighth day after pupation. All samples were frozen in liquid nitrogen and stored at -80˚C. 30 silkworms were used per biological sample, and three biological replicates were prepared and analyzed.

The BmN cell line originating from the *B. mori* ovary was used [13]. The cells were maintained in TC-100 medium (AppliChem, Gatersleben, Germany) containing 10% fetal bovine serum in 6 cm$^2$ Petri dishes at 27˚C.

### Bioinformatics analysis

To predict the functions of *BmVps13d* in *B. mori*, the amino acid sequences of Vps13d proteins in different species were searched using the BLASTP tool of the National Center for

Biotechnology Information (https://www.ncbi.nlm.nih.gov/). Conserved motifs were predicted using the InterPro server (https://www.ebi.ac.uk/interpro/). The amino acid sequences of *B. mori*, *D. melanogaster*, and *H. sapiens* Vps13d proteins were aligned using the GeneDoc software.

## RNA isolation and complementary DNA (cDNA) synthesis

The RNA was extracted from each sample using RNAiso Plus (TaKaRa, Dalian, China) according to the manufacturer's instructions and subsequently was treated with DNase I (Invitrogen, USA) to remove genomic DNA. The integrity and purity of the RNA were determined using gel electrophoresis and ultraviolet spectrophotometry. To synthesize cDNA, one microgram of total RNA was reverse transcribed using an All-in-One™miRNA First-Strand cDNA Synthesis Kit (Gene Copoeia, USA) according to the manufacturer's instructions.

## RT-qPCR analysis

The expression levels of target genes in JS and L10 were detected by RT-qPCR. The cDNA synthesized by total RNA of the midgut, silk gland, and fat body at the third day of fifth instar larval stage in JS and L10 was used as the template. The program of PCR was incubation at 95°C for 5 min, 40 cycles of 95°C for 5 s, and 60°C for 31 s, and a final dissociation. The RT-qPCR was performed in a 10 μL reaction volume that contained 1 μL template, 5 μL 2×NovoStart® SYBR qPCR SuperMix Plus (Novoprotein), 0.2 μL ROX Reference Dye II (ROX II; Novoprotein), and 0.5 μL specific primers (10 μM). *Bombyx mori glyceraldehyde-3-phosphate dehydrogenase* (*Bm GAPDH*) was used as the reference gene. The relative expression level of our target gene was calculated by using the cycle threshold value (Ct) by the method of $2^{-\Delta Ct}$, where $\Delta Ct = Ct_{\text{target gene}} - Ct_{\text{BmGAPDH}}$. The comparisons between expressions of these genes were performed with a two-tailed *t*-test: $^{*}$ $P \leq 0.05$; $^{**}$ $P \leq 0.01$; $^{***}$ $P \leq 0.001$. The primers are listed in Table 1.

## Dual luciferase assay

The dual luciferase assay system (Promega) was used to analyze the promoter activities of *BmVps13d*. By comparing *BmVps13d* sequence with that of *D. melanogaster*, the online prediction software Berkeley Drosophila Genome Project was used (http://www.fruitfly.org/seq_tools/promoter.html) to predict putative promoter sequences of *BmVps13d* and location. Different genomic fragments of *BmVps13d* containing putative promoter sequences of 2.0 kb from JS and L10 were separately subcloned into a pGL3-basic plasmid between the firefly luciferase open reading frame (ORF) and SV40 poly(A). 5 ng of the pGL3 reporter plasmid, 5 ng of the pRL-TK control plasmid mixed with 0.5μL Lipofectamine 2000 Transfection Reagent (Gibco) were transfected into the BmN cells in each well of a 96-well plate. pRL-TK vector was used to indicate transfection efficiency and used as a positive control. Equal pGL3-Basic plasmid was transfected as negative control. Cells were collected 48 h after transfection, and the relative luciferase activity was measured by normalizing the firefly luciferase level to the *Renilla*

**Table 1. Primers used for RT-qPCR experiment.**

| Primer name | Sequence (5'~3') | Tm (°C) |
| --- | --- | --- |
| *Bm GAPDH* **For** | TTCATGCCACAACTGCTACA | 60°C |
| *Bm GAPDH* **Rev** | AGTCAGCTTGCCATTAAGAG | 60°C |
| *BmVps13d* **q For** | TTCTATTGCACTTGGCCGCT | 60°C |
| *BmVps13d* **q Rev** | AATTCCACACGATCGGTCACT | 60°C |

luciferase level. The experiments were performed in triplicate with three technical repeats. Subsequently, the relative luciferase activity (firefly luciferase/renilla luciferase) was measured using a microplate reader (Synergy H1, BioTek).

### *BmVps13d* knockout by CRISPR/Cas9

Two single guide RNAs (sgRNA) were designed for *BmVps13d*. The sgRNA sites were screened according to the 5´-GG-N18-NGG-3´ rule [14]. The sgRNA sequences were predicted for potential off-target binding using CRISPR direct (http://crispr.dbcls.jp/) by performing global searches against silkworm genomic sequences [15]. All sgRNAs and oligonucleotide primer sequences for the single guide RNA (sgRNA) transgenic plasmid construction are listed in Table 2. The sgRNA sequences containing the targets were amplified, using the transgenic vector pXL[IE1-DsRed] as the initial plasmid, and then the sgRNA sequences were connected to the initial plasmid by homologous recombination to obtain the target plasmid. The single guide RNA (sgRNA) transgenic plasmid is expressed by the U6 promoter; it contains the DsRed protein gene and is initiated by IE1 to express the red fluorescent reporter gene. Plasmid map of CRISPR/Cas9 system was showed in S1 Fig. The recombinant plasmid construction was verified by PCR and sequencing. The plasmid was extracted using the QIAGEN Plasmid Midi Kit (QIAGEN GmbH, Germany), and the plasmid concentration was determined by the Nanodrop 1000 Concentration Analyzer (Thermo Fisher, USA).

### Screening of double fluorescence

The U6-sgRNA plasmid was mixed with a piggyBac helper plasmid [16] and then microinjected separately into fertilized eggs at the preblastoderm stage. $G_0$ adults were inbred, and the resulting $G_1$ progenies were screened for the single fluorescence using RFP filters (Nikon AZ100). Then the $G_1$ individuals with DsRed fluorescence were mated with Cas9 transgenic silkworms with EGFP fluorescence, and the resulting $G_2$ progenies were screened for double fluorescence to obtain the Δ*BmVps13d* mutants shown in S2 Fig.

### Genomic DNA extraction and mutagenesis analysis

Genomic PCR, followed by sequencing, was used to identify the Δ*BmVps13d* mutant individuals induced by CRISPR/Cas9 system. Genomic DNA was extracted from heterozygotes with DNA extraction buffer (1:1:2:2.5 ratio of 10% SDS to 5 mol/L NaCl to 100 mmol/L ethylenediaminetetraacetic acid to 500 mmol/L Tris-HCl, pH 8), incubated with proteinase K. Genomic DNA extraction and mutagenesis analysis and purified via a standard phenol: chloroform and isopropanol precipitation extraction, followed by RNaseA treatment. The PCR primers were listed in Table 2. The PCR products covering two target sites were cloned into the pMD19-T (TaKaRa, Dalian, China) vector and directly sequenced.

**Table 2. Primers used for RT-qPCR experiment.**

| Primer name | Sequence (5'~3') | Tm (°C) |
|---|---|---|
| Target site 1 For | TATCGTGCTCTACAAGTGAAAGATGCTCTGCGTCACTGTTTTAGAGCTAGAAATAG | 58°C |
| Target site 1 Rev | CTTATCGATACCGTCGAAAAAAAAAGCACCGACTCGGTGCC | 58°C |
| Target site 2 For | TATCGTGCTCTACAAGTGCGGACGCGCTTTCGCATGCGTTTTAGAGCTAGAAATAG | 58°C |
| Target site 2 Rev | GTTATAGATATCAGCTAGAAAAAAAAAGCACCGACTCGGTGCC | 58°C |
| Check-F | GTGGAGCTCCAGCTTTTGTT | 55°C |
| Check-R | GTGAGTCAAAATGACGCATG | 55°C |

## Investigation of the Δ*BmVps13d* mutant phenotypes

The whole life cycle of the Δ*BmVps13d* mutants was observed, and abnormal developmental characteristics were recorded, including the weights of larvae and cocoon characters. 300 silkworms of the Δ*BmVps13d* mutants and Nistari were divided into ten groups respectively, and each group was weighed every day from the first to the sixth day of fifth instar larval stage. In addition, to investigate the weights of cocoons, pupae, and cocoon shells, 120 male and female Δ*BmVps13d* mutants and Nistari were divided into 4 groups, and were weighed on the third day after cocooning. The data were statistically analyzed by using a two-tailed *t*-test.

# Results

## Characterization of *BmVps13d*

To study the functions of *BmVps13d* (SilkDB ID: BGIBMGA002843), the sequence characterizations of Vps13d proteins were obtained and compared in *B. mori*, *D. melanogaster*, and *H. sapiens* (Fig 1). Vps13d proteins in the three species all contained five domains: Chorein_N, VPS13, VPS_mid_pt, SHR-BD, and VPS13_C. Among them, Chorein_N showed mitochondrial localization. SHR-BD was a regulator of cell growth and asymmetric division. The exact functions of other domains were unknown. Besides, BmVps13d protein shared 30.84% and 34.35% identity with that of *D. melanogaster* and *H. sapiens*, respectively, which suggested that the Vps13d protein was evolutionarily conserved and might share similar biological functions in different species.

## The expression analysis of *BmVps13d*

To analyze the functions of *BmVps13d* in *B. mori* preliminarily, the expression differences of *BmVps13d* was detected in the midgut, silk gland, and fat body between JS and L10 at the third day of fifth instar larval stage by using RT-qPCR. The results showed that the expression of *BmVps13d* were significantly higher in the midgut and silk gland of JS than in that of L10 at the third day of fifth instar larval stage (Fig 2), indicating that the expression difference in the midgut might be related to absorption of nutrients, and the expression difference in silk gland might be related to silk protein.

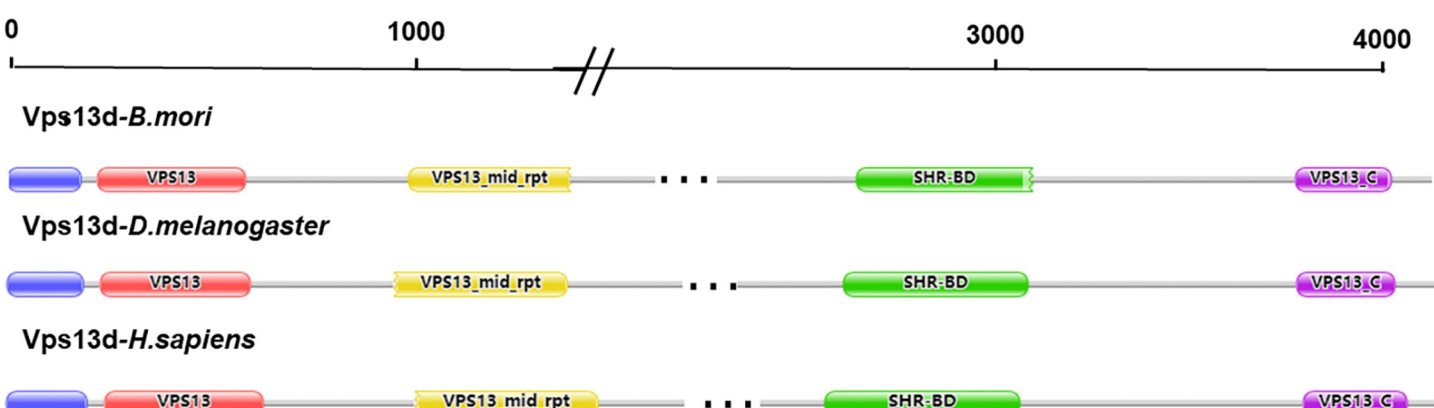

**Fig 1. Homology comparison of Vps13d proteins and the domains analysis of Vps13d proteins in *B. mori*, *D. melanogaster*, and *H. sapiens*.** The blue rectangle represents the Chorein_N domain. The red rectangle represents the VPS domain. The yellow rectangle represents the VPS_mid_pt domain. The green rectangle represents the SHR-BD domain. The purple rectangle represents the VPS13_C domain.

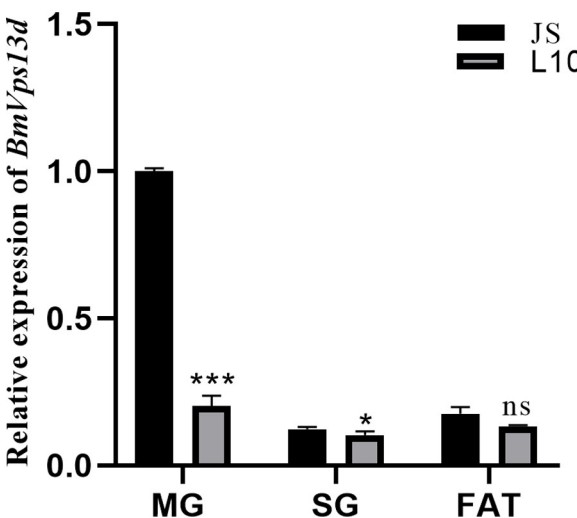

**Fig 2. *BmVps13d* expression in midgut, silk gland, and fat body of JS and L10 by RT-qPCR.** The histograms show the relative transcript levels of *BmVps13d*. MG, midgut; SG, silk gland; FAT, fat body. All data are representative of three independent experiments and are expressed as the mean ± standard error of the mean (SEM). A two-tailed *t*-test is used to evaluate gene expression difference in midgut, silk gland, and fat body between JS and L10 at the third day of fifth instar larval stage. *P*-values are described as follows: *$P \leq 0.05$, **$P \leq 0.01$, ***$P \leq 0.001$.

## The comparison of *BmVps13d* putative promoter sequences

No difference was detected in coding sequence of *BmVps13d* between JS and L10. However, relative expression of *BmVps13d* showed significant down-regulation in L10 (Fig 2), indicating that cis-regulatory elements might be responsible for *BmVps13d* down-regulation. To verify it, PCR-amplified 2.0 kb putative promoter sequences of *BmVps13d* in JS and L10, respectively. Compared to putative promoter sequences of *BmVps13d* in JS, 9 consecutive nucleotides (ATCAGAAAA) were inserted into putative promoter sequences of *BmVps13d* in L10, which was 787 bp away from the ATG. Besides, there were two deficiencies of adenine ribonucleotides in putative promoter sequences of *BmVps13d* in L10 (Fig 3). These results revealed that the differences of putative promoter sequences resulted in the different activity of *BmVps13d* promoter between JS and L10, and further affected the different expression levels of *BmVps13d* in both strains.

## Detection of *BmVps13d* promoter activity in BmN cells

To verify the differences of the *BmVps13d* promoter activity resulted from the differences of *BmVps13d* putative promoter sequences, two plasmids with different *BmVps13d* putative promoters-2.0 kb of JS and L10 were constructed respectively and transfected into dual-luciferase reporter system in BmN cells. Cells were collected for luciferase activity analysis 48 h after transfection. The pGL3-Basic plasmid was used as a control. The result showed that the activity of *BmVps13d* promoter in JS was remarkably higher than that of L10 and the control transfected with pGL3-Basic plasmid in BmN cells (Fig 4).

## The construction of the Δ*BmVps13d* mutants using CRISPR/Cas9

To determine the functions of *BmVps13d* in *B. mori* at the individual level, the sgRNA sites were designed on exon 2 and exon 4 of *BmVps13d* (Fig 5A and 5B). The target vector pXL [IE1-DsRed-U6-*Vps13d*-sgRNA] was constructed by homologous recombination, which was verified preliminarily by PCR and finally confirmed by sequencing (Fig 4C and 4D). Among

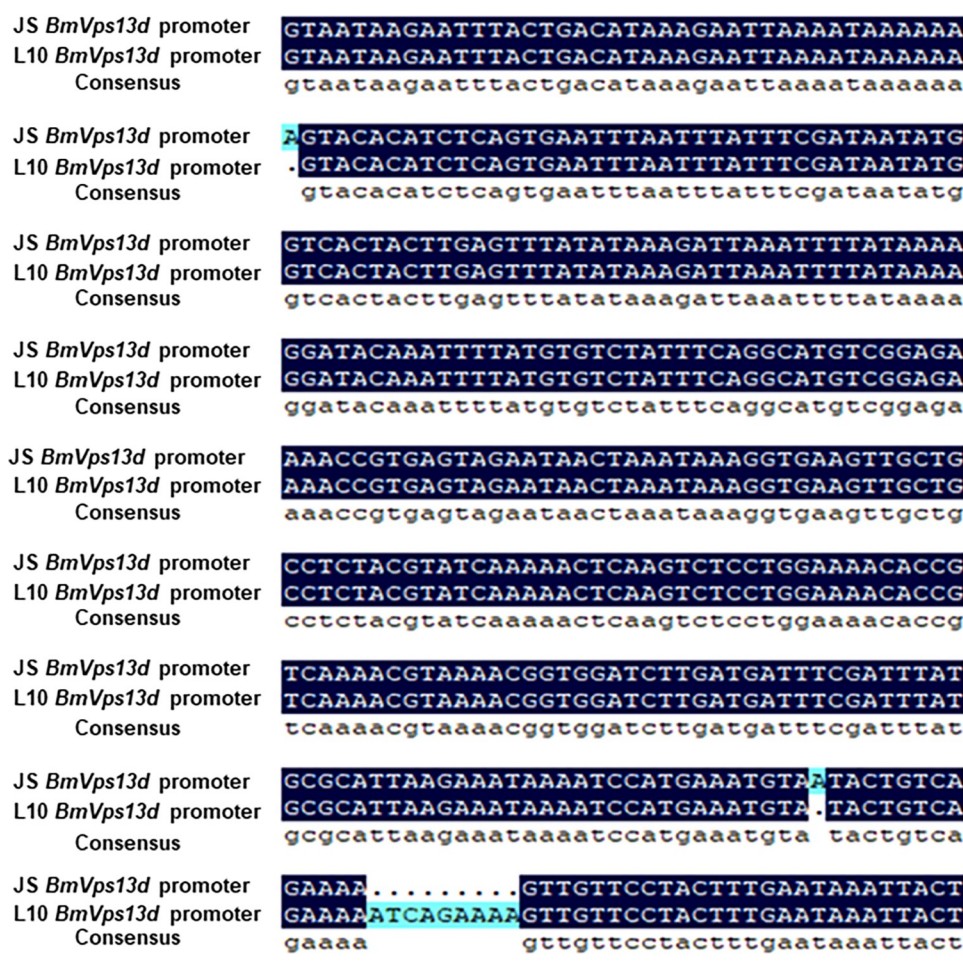

**Fig 3. Cloning and sequence comparison of putative promoters of *BmVps13d* in JS and L10 genomes.** The sequence alignment of the *BmVps13d* putative promoters-2.0 kb (2.0 kb) between JS and L10 genomes. JS *BmVps13d* promoter, the sequence of the *BmVps13d* putative promoter-2.0 kb (2.0 kb) in JS strain; L10 *BmVps13d* promoter, the sequence of the *BmVps13d* putative promoter-2.0 kb (2.0 kb) in L10 strain;---indicates that the sequence is lacking.

injected embryos of Nistari, 52% (n = 250) hatched and 85% of these individuals survived to the adult stage. The efficiency of sgRNA transgenic silkworm screened with red fluorescence was 15–25%. Finally, after the cross of *BmVps13d* sgRNA and Cas9 transgenic silkworms, the percentage of progenies with double fluorescence was 25%. To evaluate if *BmVps13d* knockout successfully, genomic DNA was extracted from two randomly-selected larvae with double fluorescence and the DNA fragment containing the target was sequenced. The result showed that we were able to generate a series of different Δ*BmVps13d* mutants (Fig 6), which could be used the individuals with double fluorescence as mutants for next experiments.

## Effect of Δ*BmVps13d* mutation on the body weight and silk yield

Δ*BmVps13d* silkworms had normal life cycles from embryo to moth. To figure out the influences on absorption of nutrients and cocoon characters, the body weights of Δ*BmVps13d* and Nistari were investigated from day 1 to 4 of the fifth instar larval stage, as well as cocoon characteristics including the weights of cocoons, pupae, and cocoon shells of the Δ*BmVps13d* mutants and Nistari on the third day after cocooning. The body weights of the Δ*BmVps13d* mutants showed a 24.7% decline than that of Nistari at the third day of fifth instar larval stage

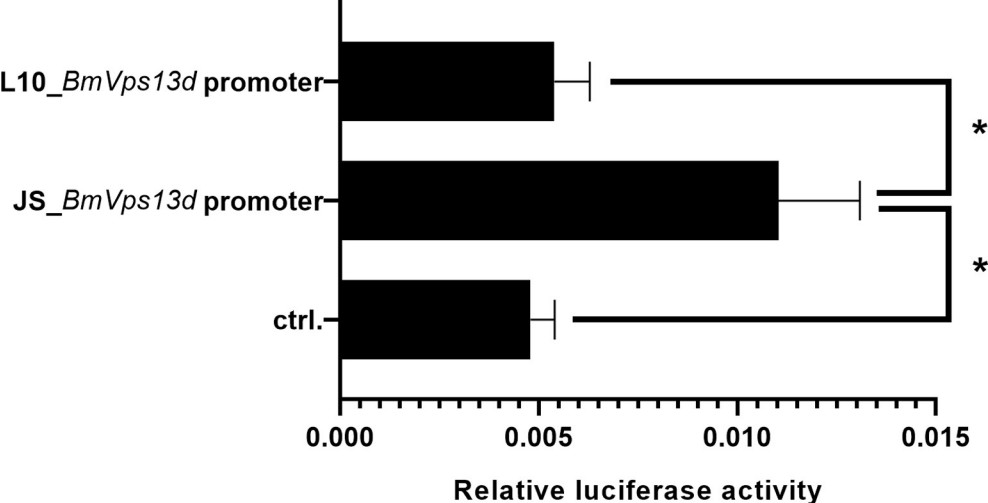

**Fig 4. *BmVps13d* promoter activity of JS is stronger than that of L10 in BmN cells.** The effect of relative luciferase activity driven by *BmVps13d* promoters of JS and L10. Ctrl., pGL3-Basic-transfected cells; L10_*BmVps13d* promoter and JS_*BmVps13d* promoter, cells transfected with *BmVps13d* promoter-2.0 kb (2.0 kb) in JS and L10 strains, respectively. All data are representative of three independent experiments. A two-tailed *t*-test is used to evaluate the difference between L10_*BmVps13d* promoter, JS_*BmVps13d* promoter, and ctrl. activity. *P*-value annotations see Fig 2.

(Fig 7A). Δ*BmVps13d* females showed 27.3%, 24.5%, and 24.8% declines in the weights of cocoons, pupae, and cocoon shells respectively; Δ*BmVps13d* males showed 11.9%, 4.3%, and 5.4% declines in the weights of cocoons, pupae, and cocoon shells respectively; females and

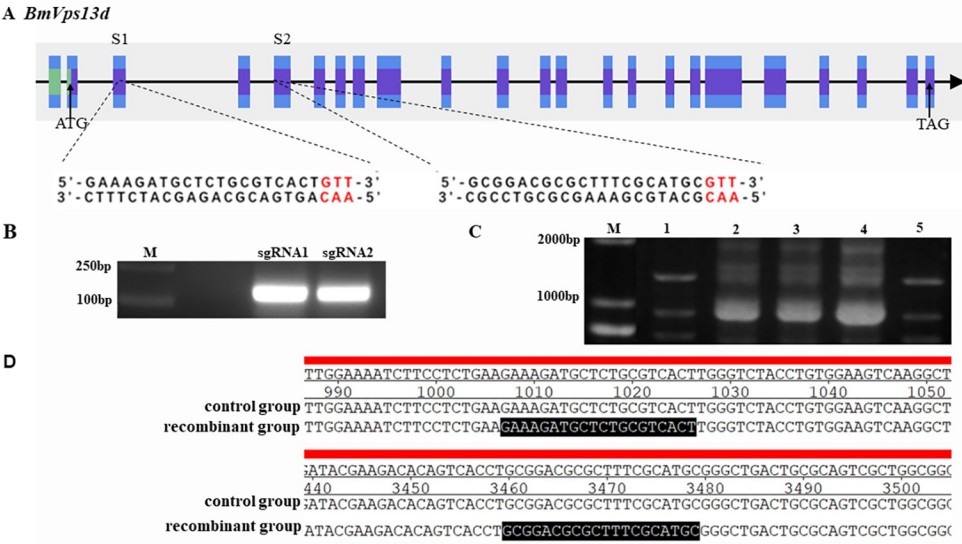

**Fig 5. Cas9/sgRNA mediated gene editing of *BmVps13d*.** (A) Schematic diagram of the sgRNA-target sites in *BmVps13d*. The purple boxes represent exons, and the black lines represent the gene locus. S1 and S2 are sgRNA sites shared in exon 2 and exon 4, respectively. The sgRNA target sequences are shown in black, and the protospacer adjacent motif (PAM) sequences are in red. (B) PCR amplification of sequences containing the sgRNA-target sites (137 bp). (C) Identification of the recombinant clones by PCR. Lane1, recombinant clone1 (1,337 bp) with sgRNA1 and sgRNA2; Lane5, recombinant clone2 (1,200 bp) with sgRNA1; Lane2-4, the failed recombinant clones. (D) Sequencing to verify the construction of the recombinant vector; the figure shows the partial sequence, the sgRNA target sites are highlighted in black.

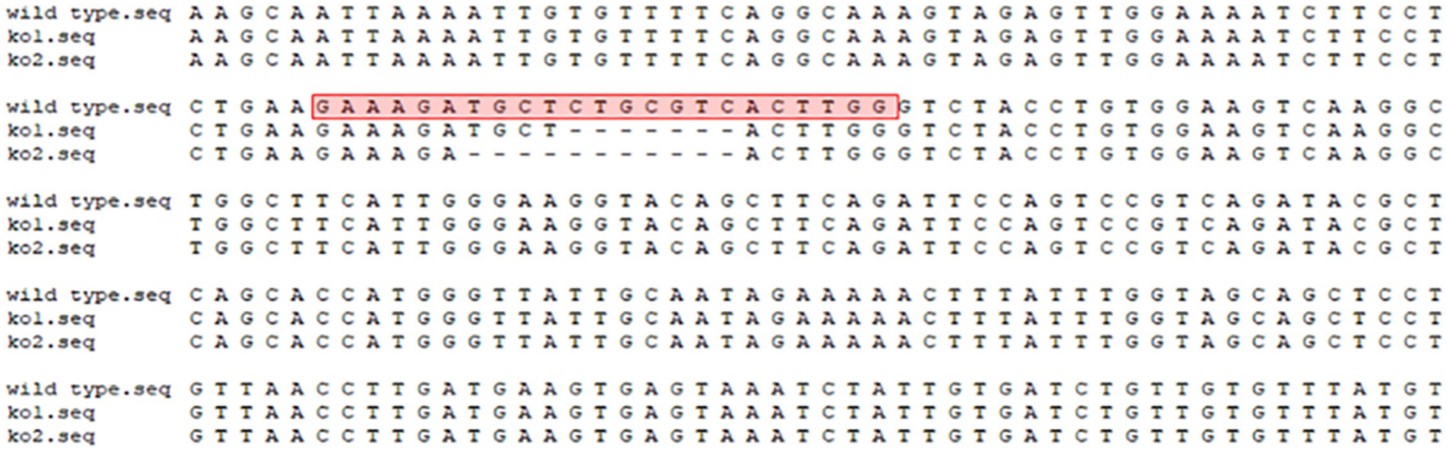

**Fig 6. CRISPR/Cas9 induces Δ*BmVps13d* mutation in *B. mori*.** Nistari is control group; ko1 is a randomly selected mutant silkworm; ko2 is another one randomly selected mutant silkworm; the target site sequence is highlighted in red;---indicates that the sequence knockout.

males of Δ*BmVps13d* showed 2.4% and 2.9% declines in the cocoon layer ratio respectively (Fig 7B).

### Effect of *BmVps13d* mutation on silk protein gene expression

To study the effect of *BmVps13d* on the expression level of silk protein genes in *B. mori*, RT-qPCR was used to detect the relative expression levels of *Fib-H*, *Fib-L*, *P25* and *Sericin-1* in silk gland of △*BmVps13d* mutant and Nistari at the third day of fifth instar larval stage respectively

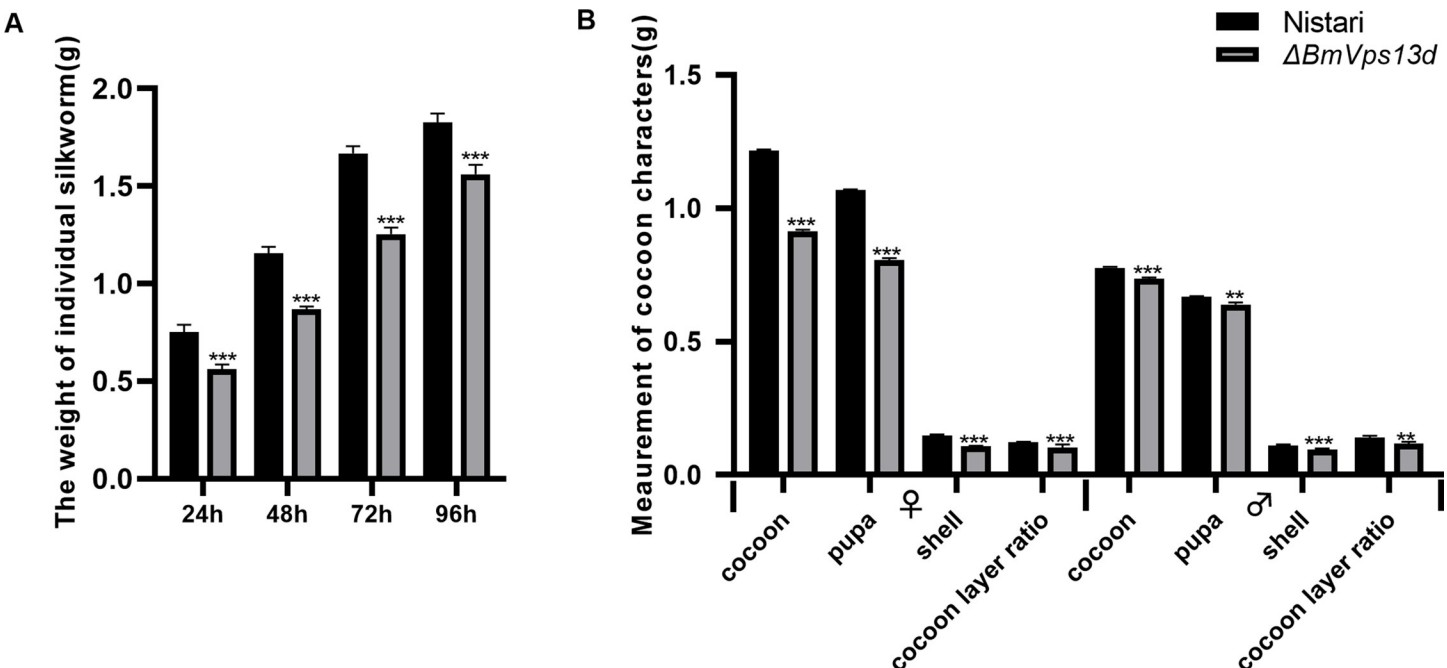

**Fig 7. Body weight and cocoon characteristic evaluation of the Δ*BmVps13d* mutants.** (A) Body weight comparisons of individual silkworms. Samples were collected every 24h from day 1 to 4 of the fifth instar larval stage. (B) Cocoon characteristic comparisons, including the weights of cocoons, pupae, and cocoon shells, Samples: Nistari (black), Δ*BmVps13d* (grey). A two-tailed *t*-test is used to evaluate the differences between the Nistari and Δ*BmVps13d*. *P*-value annotations see Fig 2.

(Fig 8). The results showed that the mutation of *BmVps13d* resulted to a significant decrease in the expression of sericin-encoding gene *Ser-1*, suggesting that *BmVps13d* had an effect on the synthesis of sericin protein.

## Discussion

The ESCRT signal pathway is composed of the peripheral membrane complexes, including ESCRT-0, -I, -II, -III, and Vps4-Vta1, and the ALIX homodimer [17]. Correlational research indicated that beyond MVB formation and endocytosis, ESCRT complexes have crucial functions in cytokinesis, and mitophagy [18–21]. Knocking down *BmVps4* resulted in silkworm metamorphosis reduction, including inhibiting silk gland growth, shortening spinning time, prolonging pupation, reducing the pupal size, and weight [22]. According to results of the bioinformatics analysis, *BmVps13d* belongs to this signal pathway and may have a similar function with *BmVps4* in *B. mori*.

It has been reported that *Vps13d* plays an important role in the ESCRT signal pathway. *Vps13d* mediate the ESCRT-dependent remodeling of lipid droplet membranes to facilitate fatty acids transfer at mitochondria-LD contacts in mammalian cells [23]. Another study demonstrates that the ubiquitin-binding protein *Vps13d* functions downstream of the fission factor Drp1 to control the mitochondrial size and autophagic clearance in *Drosophila* midgut cells. Mitophagy maintains mitochondrial homeostasis and cell health [24]. Besides, *Vps13d* is an essential gene that is necessary for the development and energy metabolism in *D. melanogaster* [11]. Combining with the results of homology comparison of Vps13d proteins (Fig 1), we speculated that *Vps13d* might have similar functions.

To preliminarily analyze the functions of *BmVps13d* in *B. mori*, we performed expression analysis. The result indicated that the expressions of *BmVps13d* were significantly higher in the midgut and silk gland of JS than in that of L10 at the third day of fifth instar larval stage (Fig 2). However, the reason for this was unclear. It's been reported that the genome sequencing analysis of different silkworm strains and the dual luciferase assay revealed that a 10 bp

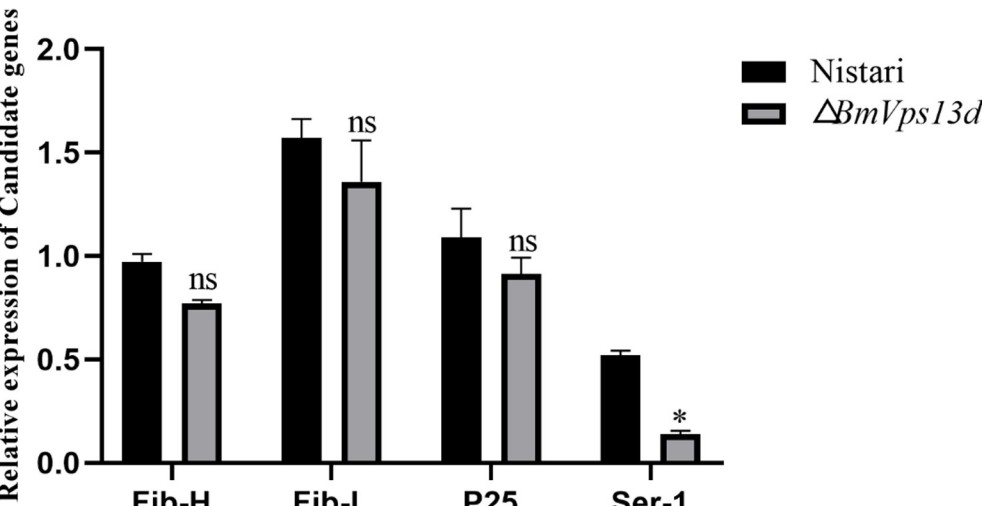

**Fig 8. The relative expression levels of silk protein genes were detected by RT-qPCR in △*BmVps13d* mutant and Nistari strain.** The histograms show the relative transcript levels of silk protein genes. Nistari (black)、△*BmVps13d* (grey). All data are representative of three independent experiments and are expressed as the mean ± standard error of the mean (SEM). A two-tailed *t*-test is used to evaluate gene expression difference in midgut, silk gland, and fat body between JS and L10 at the third day of fifth instar larval stage. *P*-values are described as follows: $^*P \leq 0.05$, $^{**}P \leq 0.01$, $^{***}P \leq 0.001$.

insertion in the putative promoter region of the *Bm-app* gene in minute wing mutant strain, which decreased *Bm-app* promoter activity [25]. Therefore, this study further analyzed putative promoter sequences of *BmVps13d*. In our study, a 9 consecutive nucleotides fragment inserted and two deficiencies of adenine ribonucleotides by found in the putative promoter of *BmVps13d* in L10, which resulted in decreasing *BmVps13d* promoter activity in L10 (Fig 3). To further verify *BmVps13d* promoter activity of JS and L10, we constructed and transfected two plasmids with the *BmVps13d* putative promoter sequences of JS and L10 into dual-luciferase reporter system in BmN cells, respectively. The result showed that the activity of *BmVps13d* promoter in JS was remarkably higher than that of the L10 and the control transfecting with pGL3-Basic plasmid in BmN cells (Fig 4). The result verified that the sequence differences of the putative promoters resulted in the different activity of *BmVps13d* promoter in JS and L10.

To further determine the functions of *BmVps13d* in *B. mori*, *BmVps13d* knockout by CRISPR/Cas9-mediated targeted mutagenesis system, which resulted in a decline in body weights, as well as the weights of cocoons, pupae, and cocoon shells in mutant silkworms (Fig 7). It was possible that *BmVps13d* expression level influenced the ESCRT signal pathway, and resulted in the development and the silk protein synthesis in mutant silkworms finally. The thumbnail image of mitochondrial autophagy mediated by *BmVps13d* in the ESCRT signal pathway was shown in S3 Fig. Besides, the ∆*BmVps13d* mutants could grow normally from larvae to adults, indicating that the inactivation of *BmVps13d* could not influence the normal development of tissues in different stages of silkworm.

## Conclusions

In conclusion, this study is first to investigated the influence of *BmVps13d* on the body wight by CRISPR/Cas9 in *B. mori*. We propose that *BmVps13d* may have a great influence on silk production. Our findings provide new insights into *BmVps13d* gene related to the development, and provide new clues for silkworm breeding with high silk yield.

## Supporting information

**S1 Fig. Plasmid map of CRISPR/Cas9 system.** A, The sgRNA expression vector with red fluorescent protein (RFP) reporter gene; B, The Cas9 expression vector with enhanced green fluorescent protein (eGFP) reporter gene.
(TIF)

**S2 Fig. Identification of positive mutant silkworm.** The picture shows the newly hatched first instar silkworm, scale bar = 1 mm. The red fluorescence, the ∆*BmVps13d* silkworm contains the sgRNA transgenic plasmid; the green fluorescence, the ∆*BmVps13d* silkworm contains the non-Cas9 transgenic plasmid.
(TIF)

**S3 Fig. The thumbnail image of the ESCRT signal pathway.** The picture shows the mechanism and process of mitochondrial autophagy and clearance mediated by *BmVps13d* in ESCRT pathway.
(TIF)

**S1 Table. The RT-qPCR original data.**
(XLSX)

**S1 Raw images.**
(PDF)

## Acknowledgments

All authors have read and agreed to the published version of the manuscript.

## Author Contributions

**Conceptualization:** Luochao Zhao.

**Data curation:** Luochao Zhao, Yunhui Kong.

**Formal analysis:** Luochao Zhao.

**Funding acquisition:** Muwang Li.

**Investigation:** Luochao Zhao.

**Methodology:** Luochao Zhao.

**Resources:** Muwang Li.

**Software:** Luochao Zhao, Sheng Qin, Yunhui Kong.

**Supervision:** Luochao Zhao, Muwang Li.

**Validation:** Muwang Li.

**Visualization:** Luochao Zhao.

**Writing – original draft:** Luochao Zhao.

**Writing – review & editing:** Luochao Zhao, Xia Sun, Xueyang Wang.

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
