## [Decision Letter · Decision Letter 0]

24 May 2022

PONE-D-22-12066Bombyx mori Vps13d is a key gene affecting the silk yieldPLOS ONE

Dear Dr. Li,

Thank you for submitting your manuscript to PLOS ONE. After careful consideration, we feel that it has merit but does not fully meet PLOS ONE’s publication criteria as it currently stands. Therefore, we invite you to submit a revised version of the manuscript that addresses the points raised during the review process.

As you can see from the attached comments, both reviewers find the manuscript interesting and novel. However, they recommend to resolve some critical issues (which they highlights in their comments) before acceptance fo publication. I suggest to revise carefully the manuscript according to the reviewers' suggestions. Please, note that the rest of this email is automatically generated.

We look forward to receiving your revised manuscript.

Kind regards,

René Massimiliano Marsano, Ph.D.

Academic Editor

PLOS ONE

Journal Requirements:

2. During your revisions, please confirm whether the wording in the title is correct and update it in the manuscript file and online submission information if needed. Specifically, we believe the title should read 'Bombyx mori Vps13d is a key gene affecting silk yield.

5.PLOS ONE now requires that authors provide the original uncropped and unadjusted images underlying all blot or gel results reported in a submission’s figures or Supporting Information files. This policy and the journal’s other requirements for blot/gel reporting and figure preparation are described in detail at https://journals.plos.org/plosone/s/figures#loc-blot-and-gel-reporting-requirements and https://journals.plos.org/plosone/s/figures#loc-preparing-figures-from-image-files. When you submit your revised manuscript, please ensure that your figures adhere fully to these guidelines and provide the original underlying images for all blot or gel data reported in your submission. See the following link for instructions on providing the original image data: https://journals.plos.org/plosone/s/figures#loc-original-images-for-blots-and-gels.

Reviewers' comments:

Reviewer's Responses to Questions

**Comments to the Author**

1. Is the manuscript technically sound, and do the data support the conclusions?

Reviewer #1: Yes

Reviewer #2: Yes

2. Has the statistical analysis been performed appropriately and rigorously? 

Reviewer #1: Yes

Reviewer #2: Yes

3. Have the authors made all data underlying the findings in their manuscript fully available?

Reviewer #1: Yes

Reviewer #2: Yes

4. Is the manuscript presented in an intelligible fashion and written in standard English?

Reviewer #1: Yes

Reviewer #2: Yes

5. Review Comments to the Author

Reviewer #1: The silk yield is an important indicator to measure the quality of silkworm varieties. This paper determined that the difference in BmVps13d gene expression between JS and L10 may be related to the difference in promoter sequence, and verified that BmVps13d gene has an important effect on silk yield of silkworm using the CRISPR/Cas9 system. The findings are interesting and meaningful. It is acceptable after some revision.

1. Why were the sequence characterizations of Vps13d proteins compared in B. mori, D. melanogaster, and H. sapiens in Figure 1?

2. Line 178-180, “The results showed that the expression of BmVps13d were significantly higher in the midgut and silk gland of JS than in that of L10 at the third day of fifth instar larval stage (Fig 2)”. However, Figure 2 does not reflect the expression of BmVps13d significantly higher in the silk gland of JS than in that of L10. It is recommended to provide original data or change the plotting method.

3. Line 252-257, “The body weights of the ΔBmVps13d mutants showed a 24.7% decline than that of Nistari at the third day of fifth instar larval stage (Fig 7 A). ΔBmVps13d females showed 27.3%, 24.5%, and 24.8% declines in the weights of cocoons, pupae, and cocoon shells respectively; ΔBmVps13d males showed 11.9%, 4.3%, and 5.4% declines in the weights of cocoons, pupae, and cocoon shells respectively; 256 females and males of ΔBmVps13d showed 2.4% and 2.9% declines in the cocoon layer ratio respectively (Fig 7 B).” It is known that the body weight of the larvae directly affects the weight of cocoons, pupae, and cocoon shells. Knockout of BmVps13d resulted in reduced larval weight, which directly affected cocoons, pupae, and cocoon shells. Therefore, a direct relationship between the BmVps13d gene and cocoons, pupae, and cocoon shells cannot be demonstrated. Therefore, the conclusion needs to be revised.

4. It is recommended that authors analyze the expression of silk protein genes in Nistari and ΔBmVps13d mutants.

Reviewer #2: 1. The promoter sequence in Figure 4 is the same as in Figure 3?

2. Page 6: lines 114-115: the role of pRL-TK vector in dual luciferase assay should be demonstrated.

3. Two sgRNA sites of BmVps13d were designed. Why was only one detected in Figure 6?

4. Page 8: lines 158-160: Why were ΔBmVps13d mutants and Nistari weighed on the third day after cocooning?

5. How about the effect of BmVps13d on the expression of silk protein genes? As an important information, the author should detect it.

6. There are some writing errors in the manuscript, please proofread the full text carefully.

6. PLOS authors have the option to publish the peer review history of their article (what does this mean?). If published, this will include your full peer review and any attached files.

Reviewer #1: No

Reviewer #2: No

---

## [Author Response · Author response to Decision Letter 0]

6 Jun 2022

Reviewer #1: The silk yield is an important indicator to measure the quality of silkworm varieties. This paper determined that the difference in BmVps13d gene expression between JS and L10 may be related to the difference in promoter sequence, and verified that BmVps13d gene has an important effect on silk yield of silkworm using the CRISPR/Cas9 system. The findings are interesting and meaningful. It is acceptable after some revision.

1.Why were the sequence characterizations of Vps13d proteins compared in B. mori, D. melanogaster, and H. sapiens in Figure 1?

The functions of Vps13d have been reported in D. melanogaster and H. sapiens, this study predicted and compared the functional domain of this gene protein in B. mori, D. melanogaster and H. sapiens.

2. Line 178-180, “The results showed that the expression of BmVps13d were significantly higher in the midgut and silk gland of JS than in that of L10 at the third day of fifth instar larval stage (Fig 2)”. However, Figure 2 does not reflect the expression of BmVps13d significantly higher in the silk gland of JS than in that of L10. It is recommended to provide original data or change the plotting method.

The original data of Figure 2 has been provided in S4_Table.

3. Line 252-257, “The body weights of the ΔBmVps13d mutants showed a 24.7% decline than that of Nistari at the third day of fifth instar larval stage (Fig 7 A). ΔBmVps13d females showed 27.3%, 24.5%, and 24.8% declines in the weights of cocoons, pupae, and cocoon shells respectively; ΔBmVps13d males showed 11.9%, 4.3%, and 5.4% declines in the weights of cocoons, pupae, and cocoon shells respectively; 256 females and males of ΔBmVps13d showed 2.4% and 2.9% declines in the cocoon layer ratio respectively (Fig 7 B).” It is known that the body weight of the larvae directly affects the weight of cocoons, pupae, and cocoon shells. Knockout of BmVps13d resulted in reduced larval weight, which directly affected cocoons, pupae, and cocoon shells. Therefore, a direct relationship between the BmVps13d gene and cocoons, pupae, and cocoon shells cannot be demonstrated. Therefore, the conclusion needs to be revised.

The conclusion has been revised in lines 307-309.

4. It is recommended that authors analyze the expression of silk protein genes in Nistari and ΔBmVps13d mutants.

The expression of silk protein genes in Nistari and ΔBmVps13d mutants have been analyzed in

Lines 263-270.

Reviewer #2: 

1. The promoter sequence in Figure 4 is the same as in Figure 3?

Yes，the promoter sequence in Figure 4 is the same as in Figure 3.

2. Page 6: lines 114-115: the role of pRL-TK vector in dual luciferase assay should be demonstrated.

The role of pRL-TK vector in dual luciferase assay has been demonstrated in lines 114.

3. Two sgRNA sites of BmVps13d were designed. Why was only one detected in Figure 6?

The target site2 missed the target and was not knocked out successfully. 

4. Page 8: lines 158-160: Why were ΔBmVps13d mutants and Nistari weighed on the third day after cocooning?

During this period, silkworms have been completely cocooned, and phenotypic investigation is most appropriate at this time.

5. How about the effect of BmVps13d on the expression of silk protein genes? As an important information, the author should detect it.

The expression of silk protein genes in Nistari and ΔBmVps13d mutants have been detected in lines 263-270.

6. There are some writing errors in the manuscript, please proofread the full text carefully.

We proofread the full text and correct some writing errors in lines 11、14、20、28、70、79、82-83、165、167-170、181、232、292、298、308、309、311.

---

## [Editor Report · Decision Letter 1]

20 Jun 2022

Bombyx mori Vps13d is a key gene affecting silk yield

PONE-D-22-12066R1

Dear Dr. Li,

We’re pleased to inform you that your manuscript has been judged scientifically suitable for publication and will be formally accepted for publication once it meets all outstanding technical requirements.

Kind regards,

René Massimiliano Marsano, Ph.D.

Academic Editor

PLOS ONE
---

## [Editor Report · Acceptance letter]

24 Jun 2022

PONE-D-22-12066R1 

*Bombyx mori Vps13d* is a key gene affecting silk yield 

Dear Dr. Li:

I'm pleased to inform you that your manuscript has been deemed suitable for publication in PLOS ONE. Congratulations! Your manuscript is now with our production department. 

Kind regards, 

on behalf of

Dr. René Massimiliano Marsano 

Academic Editor

PLOS ONE